


# Continuum model of wave propagation in fragmented media: linear damping approximation

Maxim Khudyakov[1], Arcady V. Dyskin[1], Elena Pasternak[2]

[1]School of Civil, Environmental and Mining Engineering, The University of Western Australia, Perth, 6009, Australia
[2]School of Mechanical and Chemical Engineering, The University of Western Australia, Perth, 6009, Australia

*Correspondence to*: Maxim Khudyakov (maxim.khudyakov@research.uwa.edu.au)

**Abstract.** Energy dissipation during wave propagation in fragmented geomaterials can be caused by independent movement of fragments leading to energy loss on their impact. By considering a pair of impacting fragments at times much greater than the period of their oscillations we show that at large time scale, the dynamics of the pair can be described by a linear viscous model with damping coefficient expressed through the restitution coefficient representing energy loss on impact. Wave propagation in fragmented geomaterials is also considered at the large time scale assuming that the wavelengths are much larger than the fragment sizes such that the attenuation associated with wave scattering on the fragment interfaces can be neglected. These assumptions lead to Kelvin-Voigt model of wave propagation, which allows the determination of dispersion relationship. As the attenuation and dispersion are not related to the rate dependence of rock deformation, but rather to the interaction of fragments the increasing damping and dispersion at low frequencies can be seen as an indication of fragmented nature of the geomaterial and the capacity of the fragments for independent movement.

## 1 Introduction

Geomaterials are often fragmented with the fragments covering different scales. This makes it important to understand the properties of wave propagation in such geomaterials. Fragmented materials are characterised by three major features. First is the bilinear nature of contacts when stiffness in compression is considerably higher than stiffness in tension. Bilinear oscillators feature multiple resonances, both multi-harmonic and sub-harmonic (see Dyskin et al., 2007; Dyskin et al., 2010; Dyskin et al., 2012d and literature cited there). Furthermore, chains of bilinear oscillators possess a rich structure of main resonances also accompanied by multi-harmonic and sub-harmonic ones (Shufrin et al., 2012; Dyskin et al., 2014). This structure of resonances may give an explanation to the observed spectral peaks in oscillations of blocky media (Kurlenya et al., 1996a; Kurlenya et al., 1996b; Kurlenya et al., 1996c). The effect of bilinearity on wave propagation was analysed by Kuznetsova et al. (2016).

Second is the possibility of block rotations. The bending between fragments leads to elbowing of the neighbouring fragments in the course of their mutual rotations (Pasternak et al., 2006) as well as the dependence of bending stiffness on the moments (Pasternak et al., 2012; Shufrin et al., 2014). Furthermore, non-sphericity of fragments creates in the presence of compression


an effect of apparent negative stiffness (Dyskin and Pasternak, 2012a; Dyskin and Pasternak, 2012b; Dyskin and Pasternak, 2012c). The resulting negative Cosserat shear modulus and its influence on wave propagation were analysed by Pasternak et al. (2016). It was shown that such a medium does not possess a critical frequency; subsequently the twist wave and both shear rotational waves of all frequencies can propagate.

Third is the energy dissipation associated with impact of blocks characterised by low restitution. The main feature of this type of dissipation is that it acts only at the neutral position of the oscillators formed by pairs of adjacent blocks. In this paper we consider only this special type of energy dissipation and its influence on P- and S-wave propagation assuming that otherwise the fragmented medium is unimodulus (the same stiffness in compression and tension) and neglecting the effect of rotations. We assume that the wave length is considerably larger than the fragment sizes such that energy dissipation associated with

scattering over interfaces can be neglected.

## 2 A pair of fragments with impact damping

Mathematically, the basic dynamic element involved in the process of wave propagation through fragmented geomaterials is a pair of neighbouring fragments, which can be modelled as a free undamped oscillator consisting of a single mass on undamped spring complimented by a condition that velocity decreases each time the system passes through the neutral point,

a second order differential equation (1) is analysed. Herein, the velocity reduction is governed by a restitution coefficient $\alpha \in [0,1]$.

$$\begin{cases} \ddot{x} + \omega_0^2 x = 0, \quad x(0) = x_0, \quad v(0) = \dot{x}(0) = v_0 \\ v(T_i + 0) = \alpha \cdot v(T_i - 0) \quad at \quad x(T_i) = 0 \end{cases}. \tag{1}$$

Introducing dimensionless groups $X = \dfrac{x}{l}$, $\tau = \omega_0 t$, $X_0 = \dfrac{x_0}{l}$, $V_0 = \dfrac{v_0}{l\omega_0}$, Eq. (1) can be rewritten as:

$$\begin{cases} X'' + X = 0, \quad X(0) = X_0, \quad V(0) = X'(0) = V_0 \\ V(T_i + 0) = \alpha \cdot V(T_i - 0) \quad at \quad X(T_i) = 0 \end{cases}, \tag{2}$$

The solution of Eq. (2) prior to the first impact is a commonplace:

$$X(\tau) = X_0 \cdot \cos(\tau) + V_0 \cdot \sin(\tau) = A \cdot \sin(\tau + \varphi), \tag{3}$$

where $A = \sqrt{X_0^2 + V_0^2}$, $\varphi = \arctan \dfrac{X_0}{V_0}$. $\tag{4}$

Analysing the second form of the solution, it is easy to determine the time of the first impact:





$$T_1 = \begin{cases} \pi - \arctan \dfrac{X_0}{V_0}, & if \ \dfrac{X_0}{V_0} \geq 0 \\[2mm] -\arctan \dfrac{X_0}{V_0}, & if \ \dfrac{X_0}{V_0} < 0 \end{cases}. \tag{5}$$

After the first impact, the velocity of oscillations reduces by restitution coefficient $\alpha$. Thus, a subsequent solution has the following form:

$$X(\tau) = \int \alpha^{H(\tau - T_1)} \cdot A \cdot \cos(\tau + \varphi) d\tau = \alpha^{H(\tau - T_1)} \cdot A \cdot \sin(\tau + \varphi), \tag{6}$$

where $H(t)$ is the Heaviside function.

In this system, each next impact starting from the second occurs after time $\pi$ from the previous impact decreasing stepwise the amplitude by $\alpha$; therefore, the general solution of the system is

$$X(\tau) = \alpha^{H(\tau - T_1) + H(\tau - (T_1 + \pi)) + \ldots + H(\tau - (T_1 + n\pi)) + \ldots} \cdot A \cdot \sin(\tau + \varphi) = \alpha^{\sum\limits_{m=0}^{\infty} H(\tau - (T_1 + m\pi))} \cdot A \cdot \sin(\tau + \varphi). \tag{7}$$

It should be noted that the analysed problem is somewhat similar to a problem of a bilinear oscillator with an infinite stiffness

in one direction (Dyskin et al., 2012d; Dyskin et al., 2013; Guzek et al., 2016) or a ball bouncing off a solid wall (Luck& Mehta, 1993; Anagnostopoulos, 2004; Jankowski, 2006), with the same coefficient of restitution, in terms of the amplitude. The main difference is that, described by Eq. (1), the range of $\alpha$ for the former type of a problem lies in a negative region, between -1 and 0, which leads to function $X$ being only in a positive domain. As a result, for equal and physically admissible boundary conditions and absolute values $\alpha$, the odd half cycles of the solutions for those two types of problems are identical

and the even half cycles are symmetrical about the axis $X = 0$. Both of these types of problems are demonstrated in Fig. 1.

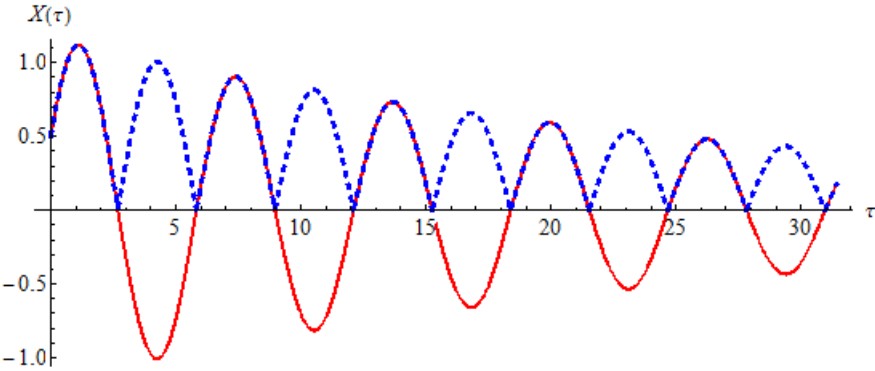

**Figure 1: Vibrations for fragmented media (red solid line) and impact oscillator (blue dashed line) for absolute values of $\alpha = 0.9$ and initial conditions $X_0 = 0.5$ and $V_0 = 1.0$.**





## 3 Equivalent linear damping

Although the analysed model is relatively simple to describe, its implementation for wave propagation problems is challenging, the reason being that the boundaries of the time intervals where the system behaves linearly are not known a priori as they are influenced by incomplete restitution and hence need to be determined step by step. Therefore, an equivalent continuous model,

e.g., Kelvin-Voigt model, with effective coefficients should be chosen and the relationship between it and the original model should be established.

Kelvin-Voigt model represents a free oscillation of a mass with damping that can be characterised in a dimensionless form by the damping coefficient $\zeta > 0$:

$$Y'' + 2 \cdot \zeta \cdot Y' + Y = 0, \ Y(0) = X_0, \ Y'(0) = V_0. \tag{8}$$

Equation (8) has three different types of solutions, depending on the value of $\zeta$, which are overdamped ($\zeta > 1$), critically damped ($\zeta = 1$) or underdamped ($0 < \zeta < 1$):

$$Y(\tau_2) = \begin{cases} \exp(-\zeta\tau_2) \cdot A_2 \cdot \sin\left(\tau_2\sqrt{1-\zeta^2} + \varphi_2\right), \ if \ 0 < \zeta < 1 \\ \exp(-\tau_2) \cdot \left(X_0 + (X_0 + V_0)\tau_2\right), \ if \ \zeta = 1 \\ \frac{1}{2} \cdot \exp(-\zeta\tau_2) \cdot \left(\left(X_0 + \frac{V_0 + \zeta X_0}{\sqrt{\zeta^2-1}}\right) \cdot \exp\left(\tau_2\sqrt{\zeta^2-1}\right) + \left(X_0 - \frac{V_0 + \zeta X_0}{\sqrt{\zeta^2-1}}\right) \cdot \exp\left(-\tau_2\sqrt{\zeta^2-1}\right)\right), \ if \ \zeta > 1 \end{cases} \tag{9}$$

where $A_2 = \sqrt{\dfrac{X_0^2 + V_0^2 + 2X_0V_0\zeta}{1-\zeta^2}}$, $\varphi_2 = \arctan\dfrac{X_0\sqrt{1-\zeta^2}}{V_0 + X_0\zeta}$.

Among those three solution types, only the underdamped solution is physically admissible for a comparison with the proposed

model because the other two types do not intersect the axis $X = 0$. Consequently, the relationship between the damping coefficient and restitution coefficient is carried out using the underdamped solution.

Comparing the expressions inside the *sin* functions in (6) and (9) one can see that:

$$\tau_2 = \frac{\tau}{\sqrt{1-\zeta^2}} . \tag{10}$$

In the discrete model, the initial conditions consist of zero displacement and given initial velocity. So, hereafter, the initial

displacement $X_0$ is set as 0, which leads to $T_1 = \pi$.

Thus, using Eq. (10) for the Kelvin-Voigt model, it is possible to find a relation between $X(\tau)$ and $Y(\tau_2)$. For the former system, after a number of cycles $N$, with the passing time $\tau = 2N\pi$, the solution becomes:

$$X(\tau) = \alpha^{2N} \cdot V_0 \cdot \sin(2N\pi) \tag{11}$$

On the other hand, for the Kelvin-Voigt model, using Eq. (10) one has:





$$Y\left(\frac{\tau}{\sqrt{1-\zeta^2}}\right) = \exp\left(-\frac{2N\pi}{\sqrt{1-\zeta^2}}\cdot\zeta\right)\cdot\frac{V_0}{\sqrt{1-\zeta^2}}\cdot\sin\left(2N\pi\right). \qquad (12)$$

As long as the third terms in Eqs. (11) and (12) are identical, the relationship between the damping parameters can be determined by comparison of the amplitudes of the systems:

$$\alpha = \left(1-\zeta^2\right)^{-\frac{1}{4N}}\cdot\exp\left(-\frac{\pi\zeta}{\sqrt{1-\zeta^2}}\right). \qquad (13)$$

The first term of the right side in Eq. (13) goes rapidly to unity even for significant values of $\zeta$. Thus, for a purpose of homogenisation of the proposed model, this term can be substituted by unity, Fig. 2:

$$\alpha = \exp\left(-\frac{\pi\zeta}{\sqrt{1-\zeta^2}}\right). \qquad (14)$$

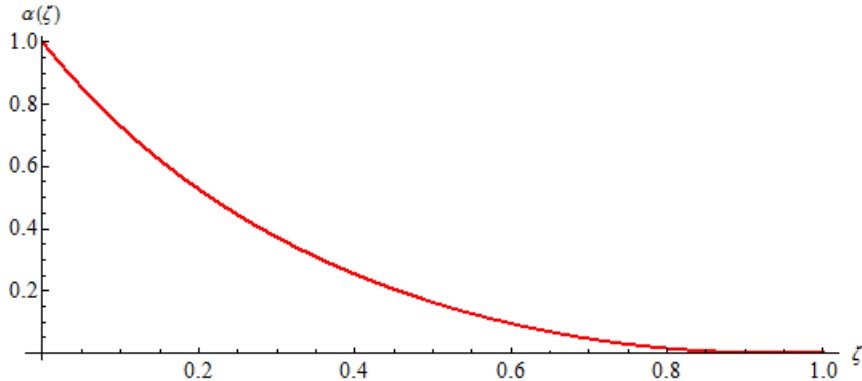

**Figure 2: Relationship (14) between $\alpha$ and $\zeta$.**

It is seen from Eq. (14) that as $\zeta = 0$, $\alpha = 1$ (the left boundary) and when $\zeta \to 1$, $\alpha \to 0$ (the right boundary). The left boundary proves that the solutions for undamped oscillations (Eq. (8), $\zeta = 0$) and oscillations with full restitution (Eq. (2), $\alpha = 1$) are identical. The right boundary corresponds to zero restitution and can be modelled by this relationship as a system

with the critical damping ($\zeta = 1$) intersecting $X = 0$ as time tends to infinity. Also, Eq. (14) shows that the relationship between the damping parameters does not depend on the initial velocity of the systems or on the current time for all $\zeta$ within the range of the underdamped oscillations. Hence, taking into consideration the previous statements, Eq. (14) should be applied for $0 \leq \alpha \leq 1$. As a result, systems with both damping and incomplete restitution can be analysed by using equivalent viscous damping, reducing the complexity of the discrete problem.



When both types of energy dissipation take place, the restitution coefficient should be replaced by a damping coefficient; therefore, a reverse relationship is also important to define. It can be found relatively easy from Eq. (14), selecting only positive roots.

$$\zeta = \ln\left(\frac{1}{\alpha}\right) \cdot \left(\pi^2 + \left(\ln\left(\frac{1}{\alpha}\right)\right)^2\right)^{-1}. \tag{15}$$

In order to analyse the accuracy of Eq. (14), an example of vibration amplitudes of an oscillator representing fragmented

media, $X(\tau)$, with different $\alpha$ and vibrations of an equivalent Kelvin-Voigt model, $Y\left(\dfrac{\tau}{\sqrt{1-\zeta^2}}\right)$, is presented in Fig. 3. The

initial conditions for both cases are $X_0 = 0.0$ and $V_0 = 1.0$.

It is seen that between the impacts, the functions can be quite different even for high values of $\alpha$. Indeed, the energy dissipation in the original system occurs at discrete times. Replacing the discrete system with a time-continuous system is

equivalent to using the time scale considerably larger than the period. At this scale, the resulting damping is the same as in the original discrete system.

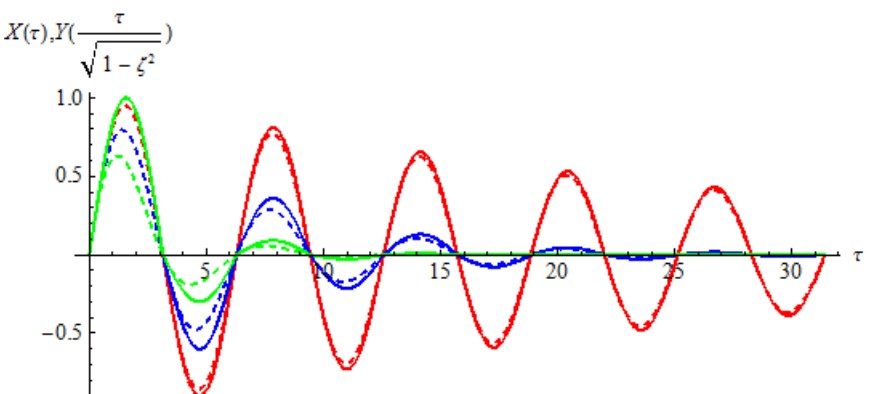

**Figure 3: Vibrations for fragmented media (solid lines) and equivalent Kelvin-Voigt model (dashed lines) for α = 0.9 (red),**
**α = 0.6 (blue), α = 0.3 (green) and initial conditions $X_0$ = 0.0 and $V_0$ = 1.0.**

The dissipated energy of vibrations with the same parameters are given for both cases by the following equations, where $W_x$ is the energy dissipation function for the discrete model and $W_y$ is the same function for equivalent Kelvin-Voigt model. These dependencies are shown in Fig. 4.

$$W_X(\tau) = \frac{1-\alpha^2}{2}\left(\sum_{m=0}^{N}\alpha^{2m} \cdot H(\tau - m\pi) \cdot V_0^2\right), \tag{16}$$

$$W_Y\left(\frac{\tau}{\sqrt{1-\zeta^2}}\right) = \frac{1}{2} \cdot \left(\frac{V_0}{\sqrt{1-\zeta^2}}\right)^2 \cdot \left(1-\zeta^2 - \exp\left(-\frac{2\tau\zeta}{\sqrt{1-\zeta^2}}\right) \cdot \left(1-\zeta^2\cos(2\tau) - \zeta\sqrt{1-\zeta^2}\sin(2\tau)\right)\right). \tag{17}$$





The dissipated energy $W_X(\tau)$, cannot be approximated by $W_Y\left(\dfrac{\tau}{\sqrt{1-\zeta^2}}\right)$ between impacts, especially for small restitution

coefficients, because here a continuous function is approximated by a stepwise function. Nevertheless, they approach the same

values at impacts; therefore, the proposed function and the equivalent continuous function (linear damping) can be used as an

approximation of the discrete one for times considerably higher than the period of free vibrations.

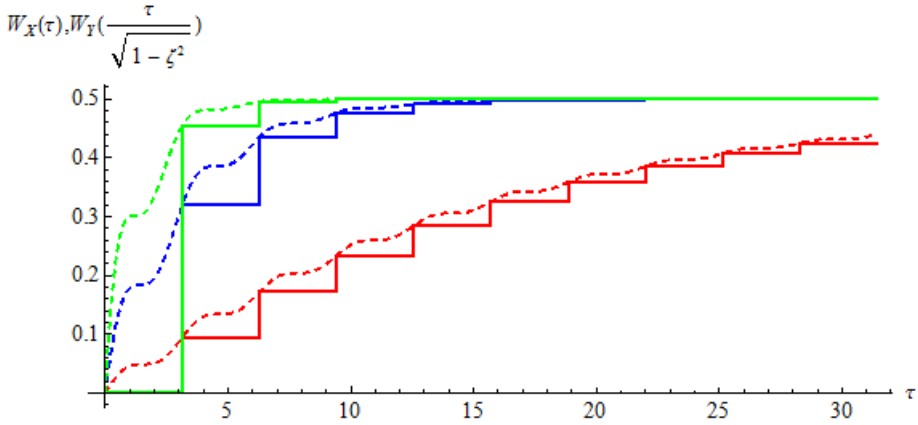

**Figure 4: Dissipated energy of vibrations for fragmented media (solid line) and equivalent Kelvin-Voigt model (dashed line) for $\alpha = 0.9$ (red), $\alpha = 0.6$ (blue), $\alpha = 0.3$ (green) and initial conditions $X_0 = 0.0$ and $V_0 = 1.0$.**

**4 Wave propagation in isotropic medium with equivalent damping**

Now, after establishing the large time scale equivalence of the discrete and continuous dynamics of a pair of fragments, the

10  wave propagation in fragmented geomaterials can be modelled by replacing the fragmented geomaterial with a visco-elastic

continuum where the energy dissipation on impact is described by a Kelvin-Voigt model. The P-wave velocity $c_p$, and

coefficient of absorption $a_p$ are expressed by the following equations (White, 1983) and are shown in Fig. 5 for different $\alpha$

$$\left(\frac{c_p}{c}\right)^2 = 2\cdot\left(1+\left(\frac{2\cdot\omega_1\cdot\ln\left(\frac{1}{\alpha}\right)}{\pi^2+\left(\ln\left(\frac{1}{\alpha}\right)\right)^2}\right)^2\right)\cdot\left(1+\sqrt{1+\left(\frac{2\cdot\omega_1\cdot\ln\left(\frac{1}{\alpha}\right)}{\pi^2+\left(\ln\left(\frac{1}{\alpha}\right)\right)^2}\right)^2}\right)^{-1}, \tag{16}$$

$$\left(a_p\cdot c\right)^2 = \left(\frac{\pi^2+\left(\ln\left(\frac{1}{\alpha}\right)\right)^2}{2\cdot\ln\left(\frac{1}{\alpha}\right)}\right)^2\cdot\left(\frac{2\cdot\omega_1\cdot\ln\left(\frac{1}{\alpha}\right)}{\pi^2+\left(\ln\left(\frac{1}{\alpha}\right)\right)^2}\right)^4\cdot\left(2\cdot\left(1+\left(\frac{2\cdot\omega_1\cdot\ln\left(\frac{1}{\alpha}\right)}{\pi^2+\left(\ln\left(\frac{1}{\alpha}\right)\right)^2}\right)^2\right)\cdot\left(\sqrt{1+\left(\frac{2\cdot\omega_1\cdot\ln\left(\frac{1}{\alpha}\right)}{\pi^2+\left(\ln\left(\frac{1}{\alpha}\right)\right)^2}\right)^2}+1\right)\right)^{-1}, \tag{17}$$



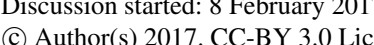


where $c$ is the P-wave velocity without damping and $\omega_1$ is wave frequency.

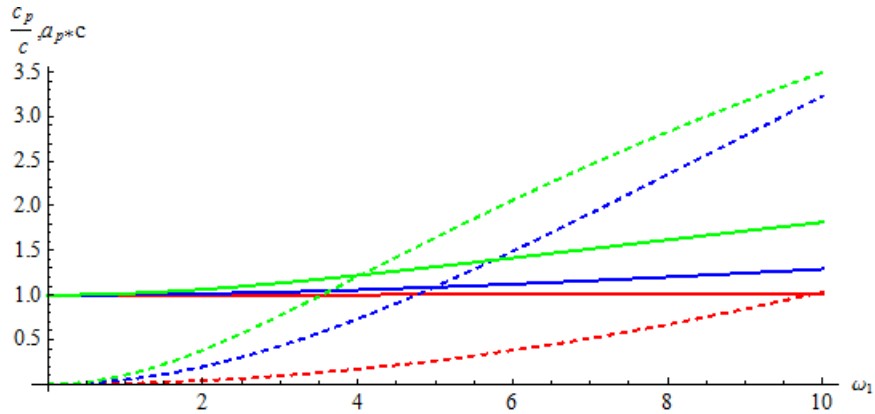

**Figure 5: Velocity characteristics of a p-wave propagating in fragmented media: phase velocity (solid line) and coefficient of absorption (dashed line) for $\alpha = 0.9$ (red), $\alpha = 0.6$ (blue), $\alpha = 0.3$ (green).**

It is seen that both the wave velocity and absorption increase with frequency, however the increase in the wave velocity becomes weaker as the restitution coefficient increases. Subsequently, the dispersion vanishes as the restitution coefficient tends to 1, i.e. the impacts are not accompanied by energy loss. It is noteworthy that these formulae can be implemented for S-waves as well.

**Conclusion**

A possible mechanism of wave attenuation in fragmented geomaterials with fragment sizes much smaller than the wavelengths is the energy loss on impact of the contacting fragment to each other. The energy loss is characterised by the restitution coefficient. It is shown that the wave propagation in such a discrete material can be modelled by wave propagation in an equivalent visco-elastic continuum if the characteristic times involved are considerably greater than the periods of oscillations of all neighbouring pairs of fragments. The attenuation is modelled by Kelvin-Voigt model; its equivalent damping being expressed through the restitution coefficient and the period of oscillations of contacting fragments averaged over all pairs. For all restitution coefficients smaller than 1, the wave velocity shows dispersion relationship, which is the stronger the smaller the restitution is. The attenuation and dispersion are not related to rate dependent rock deformation, but rather to the interaction of fragments. For that reason the effect is long-wave. Therefore, increasing damping and dispersion at low frequencies can be seen as an indication of fragmented nature of the geomaterial and the capacity of the fragments for independent movement.

**Acknowledgement**

The first author gratefully acknowledges the scholarship support from the University of Western Australia (RTP).

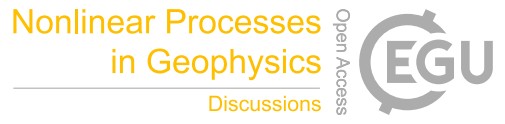



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
