# Peer review of "Continuum model of wave propagation in fragmented media: linear damping approximation"

_Nonlinear Processes in Geophysics, 2017_

## Referee Comment (RC1) · Anonymous Referee #1 · 12 Mar 2017

The paper addresses the continuum model of wave propagation in fragmented media. The obtained results look interesting, new and of some significance in mechanical engineering. The paper in its major part fits current trends and standards for international scientific publications. Its presentation is clear and concise, includes reasonable number of figures of acceptable quality. The title fairly presents the main idea of the article, yet a few shortcomings as to its narration, English and context were found. On balance, whilst major part of the material can be appropriate for publication, additional work is still needed. In my point of view, the extent of the required changes can be considered as "minor to moderate revision". Detailed concerns follow.

–>THE VOIGHT MODEL. In general, any use of simple models is subject to serious

grounding. Although the paper includes a lot of speculation and very good and useful arguments in favor of the Voight model and its applicability, the narration is sometimes puzzling. The Abstract reads: "These assumptions lead to Kelvin-Voight model of wave propagation, . . .", which makes a feeling that the use of this very model will be substantially grounded after. At the same time, introduction of the model appears at p. 4 l.5 as "e.g., Kelvin-Voight model, with . . . ", which gives a feeling that it's just a choice from a long row of models applicable to the problem. And then nothing is said about other alternatives! By the way, the paper is silent about "uniform distribution of strain" - an assumption important for applicability of the Voight model, contrary to the Maxwell model assuming a "uniform distribution of stress". Nothing is said about more extended capabilities provided by the Zener model. Testing the whole set of available models is obviously beyond the scope of a single paper, but due consideration of their basics from the point of view of their applicability is a must. I would also encourage the authors to be more cautious with terms: e.g. contrary to the wording from the above cited sentence of Abstract, the Voight model is basically NOT the "wave propagation model", but is a helpful means for modeling viscoelasticity under certain conditions. This closing remark smoothly leads us to

–> THE USE OF ENGLISH. English is mostly OK except for some awkward sentences. Please proofread and perform the grammar check carefully. Just a few examples: (i) in the last sentence of p.1 the modifier of manner "in the presence of . . ." comes between the transitive verb "to create" and its direct object "an effect . . .", which makes it hard to read, (ii) "therein" would look much better than "there" in l.22 p.1, (iii) the use of "increasing damping" in l.15 of p.1 and then in similar cases is confusing in my point of view, (iv) the sentence "Consequently, . . .." around l.15 p.4 lacks any expression of necessity I would expect (otherwise, I can't grasp the meaning). In general, I wouldn't recommend pricey editorial services, and encourage being more conservative and careful with the use of English.

–> PLEASE DELETE MISPRINTS!!! E.g.: Where are derivatives in the second order

differential equation (1) and boundary conditions p.2 l.17?

---

## Referee Comment (RC2) · Anonymous Referee #2 · 17 Mar 2017

Review of NPG-2017-3, by Maxim Khudyakov, Arcady V. Dyskin, and Elena Pasternak, "Continuum model of wave propagation in fragmented media: linear damping approximation"

The authors have presented a model of wave dissipation at wave propagation in a fragmented medium with accounting of impacts. This problem is of interest for researchers in the area of wave propagation in geomaterials. The paper is worth to be published after taking into account the items that follow. 1. Please give the estimates of wave frequencies and characteristic time of impact and define concretely the scopes of the model. 2. Kelvin – Voigt model is model of a medium, not for wave propagation. Moreover, as is generally known, this model does not satisfy the

causality condition, or the fundamental Kramers–Kronig relations [Aki K. and Richards P.G. Quantitative Seismology. Theory and Methods, Vol.1. W.H. Freeman, New York. 1980]. Please add a comment. 3. Please emphasize the differences with the literature [Anagnostopoulos, S.A. Equivalent viscous damping for modeling inelastic impacts in earthquake pounding problems. Earthquake Engineering & Structural Dynamics, 33(8), 897–902., doi:10.1002/eqe.377, 2004], particularly touching Eq. 14 and Fig. 2. 4. There is no necessity to give all the solutions of Eq. (8). The first equation of Eq. (9) is sufficient. 5. I would recommend the authors to pay attention to the problems concerning the correct accounting of the energy dissipation at impacts as in [Hunt, K. H., & Crossley, F. R. E. (1975). Coefficient of Restitution Interpreted as Damping in Vibroimpact. Journal of Applied Mechanics, 42, 440.]

Please also note the supplement to this comment:
http://www.nonlin-processes-geophys-discuss.net/npg-2017-3/npg-2017-3-RC2-supplement.pdf

———————————————————

---

## Author Comment (AC1) · 18 May 2017

**Answers on the review of Anonymous Referee #1**

The authors are grateful for the perusal and important suggestions provided by the reviewer. All the suggestions have been taken into account and the manuscript has been changed accordingly.

–>THE VOIGHT MODEL. In general, any use of simple models is subject to serious grounding. Although the paper includes a lot of speculation and very good and useful arguments in favor of the Voight model and its applicability, the narration is sometimes puzzling. The Abstract reads: "These assumptions lead to Kelvin-Voight model of wave propagation, . . .", which makes a feeling that the use of this very model will be substantially grounded after. At the same time, introduction of the model appears at p. 4 l.5 as "e.g., Kelvin-Voight model, with . . . ", which gives a feeling that it's just a choice from a long row of models applicable to the problem. And then nothing is said about other alternatives! By the way, the paper is silent about "uniform distribution of strain" - an assumption important for applicability of the Voight model, contrary to the Maxwell model assuming a "uniform distribution of stress". Nothing is said about more extended capabilities provided by the Zener model. Testing the whole set of available models is obviously beyond the scope of a single paper, but due consideration of their basics from the point of view of their applicability is a must. I would also encourage the authors to be more cautious with terms: e.g. contrary to the wording from the above cited sentence of Abstract, the Voight model is basically NOT the "wave propagation model", but is a helpful means for modeling viscoelasticity under certain conditions. This closing remark smoothly leads us to

**The authors would like to thank you very much for the remark about "wave propagation model". Kelvin-Voigt model is used in this paper to analyse energy dissipation during wave propagation. That has been changed throughout the paper. Regarding the selection of this model instead of others, e.g. Maxwell and Zener models, the explanation has been provided in the text, p.4. l.7.**

–> THE USE OF ENGLISH. English is mostly OK except for some awkward sentences. Please proofread and perform the grammar check carefully. Just a few examples: (i) in the last sentence of p.1 the modifier of manner "in the presence of . . ." comes between the transitive verb "to create" and its direct object "an effect . . .", which makes it hard to read, (ii) "therein" would look much better than "there" in l.22 p.1, (iii) the use of "increasing damping" in l.15 of p.1 and then in similar cases is confusing in my point of view, (iv) the sentence "Consequently, . . .." around l.15 p.4 lacks any expression of necessity I would expect (otherwise, I can't grasp the meaning). In general, I wouldn't recommend pricey editorial services, and encourage being more conservative and careful with the use of English.

**(i) It has been changed**

**(ii) It has been changed**

**(iii) It has been changed**

**(iv) The word "Consequently" is needed to connect the sentence providing some explanation why the underdamped solution is appropriate and the sentence stating that it used for establishing the relationship between the damping coefficient and restitution coefficient.**

–> PLEASE DELETE MISPRINTS!!! E.g.: Where are derivatives in the second order differential equation (1) and boundary conditions p.2 l.17?

The derivatives are represented by dots upon the function "*x*". One and two dots show the first and second derivatives, respectively. There has been also added a formula after Eq. (2) showing how to transform $X'$ to $\dot{x}$ and vice versa.

---

## Author Comment (AC2) · 18 May 2017

**Answers on the review of Anonymous Referee #2**

The authors are grateful for the perusal and important suggestions provided by the reviewer. All the suggestions have been taken into account and the manuscript has been changed accordingly.

1. Please give the estimates of wave frequencies and characteristic time of impact and define concretely the scopes of the model.

**This article covers just a topic of energy dissipation during wave propagation in an infinite body, so no impacts are considered.**

2. Kelvin – Voigt model is model of a medium, not for wave propagation. Moreover, as is generally known, this model does not satisfy the causality condition, or the fundamental Kramers–Kronig relations [Aki K. and Richards P.G. Quantitative Seismology. Theory and Methods, Vol.1. W.H. Freeman, New York. 1980]. Please add a comment.

**The authors would like to thank you very much for the remark about "wave propagation". Kelvin-Voigt model is used in this paper to analyse energy dissipation during wave propagation. That has been changed throughout the paper.**

3. Please emphasize the differences with the literature [Anagnostopoulos, S.A. Equivalent viscous damping for modeling inelastic impacts in earthquake pounding problems. Earthquake Engineering & Structural Dynamics, 33(8), 897–902., doi:10.1002/eqe.377, 2004], particularly touching Eq. 14 and Fig. 2.

**The main difference from the cited paper is that in our model, loss of energy occurs only at the neutral point; therefore, the equivalent linear damping function can be used only for times considerably higher than the period of free vibrations. On the other hand, in the cited article damping happens constantly during the impact of two bodies and each impact can be represented by an equivalent viscous damping.**

4. There is no necessity to give all the solutions of Eq. (8). The first equation of Eq. (9) is sufficient.

**They were included for completeness.**

5. I would recommend the authors to pay attention to the problems concerning the correct accounting of the energy dissipation at impacts as in [Hunt, K. H., & Crossley, F. R. E. (1975). Coefficient of Restitution Interpreted as Damping in Vibroimpact. Journal of Applied Mechanics, 42, 440.]

**Yes, we calculate the loss of energy by the same operation and the Eq. (16) covers all impacts simultaneously.**